# Refining soil nutrient assessment: Incorporating land use boundaries for precision agriculture

**Quan Xu** *, **Junling He***

China Geological Survey Urumqi Comprehensive Survey Center on Natural Resources, Urumqi, China

* xuquan@mail.cgs.gov.cn (QX); hejunling@mail.cgs.gov.cn (JH)

## Abstract

Soil nutrient levels play a crucial role in determining crop yield. A comprehensive understanding of the spatial distribution patterns and evaluation grades of soil nutrients is of significant practical importance for informed fertilization practices, enhancing crop production, and optimizing agricultural land utilization. This study focuses on the urban area of Kashi Prefecture in Xinjiang as a case study. Utilizing soil sample data, GIS spatial interpolation analysis was conducted, incorporating plot boundary information to propose a comprehensive evaluation method for assessing soil nutrient levels at the plot level. Experimental findings revealed the following: (1) The average values of soil organic matter (SOM), total nitrogen (AN), total potassium (AK), and total phosphorus (AP) in the study area were determined to be 13.3 g/kg, 0.74 g/kg, 0.33 g/kg, and 0.03 g/kg, respectively. Among these, AN and SOM were classified as the fourth grade, indicating relatively deficient levels, while AK and AP were classified as the first and second grade, indicating relatively abundant levels. (2) The comprehensive evaluation of soil nutrient grades in the study area primarily fell within the third, fourth, and second grades, representing areas of 29.08 km$^2$, 25 km$^2$, and 4.05 km$^2$, accounting for 50.03%, 43%, and 6.97% of the total area, respectively. (3) The evaluation results of soil nutrient levels at the plot level emphasized the boundary characteristics and provided a more refined assessment grade. This evaluation method is better suited to meet the practical production requirements of farmers and is considered feasible. The outcomes of this study can serve as a reference for precision agriculture management.

## Introduction

Soil nutrients play a crucial role in determining crop yield, and conducting a comprehensive evaluation of soil nutrients is essential for guiding farmers towards scientific fertilization practices, maintaining land productivity, promoting crop yield increase, and fostering sustainable development in the agricultural economy [1–3]. By combining traditional mathematical statistical methods with Geographic Information System (GIS) technology, researchers can effectively study the spatial distribution characteristics and evolutionary patterns of soil nutrients at

**Data Availability Statement:** The data are available from the Open Science Framework database. https://osf.io/a9ezj/ DOI 10.17605/OSF.IO/A9EZJ.

**Funding:** Quan Xu was supported by the Science and Technology Innovation Foundation of the Command Center for Comprehensive Survey of

Natural Resources (KC20230015), the Geological Survey of China (DD20230740), and the National Natural Science Foundation of China (U2003109). He played a role in data collection, methodology, and writing – original draft. Junling He was supported by the Science and Technology Innovation Fund of the Command Center for Integrated Survey of Natural Resources (KC20220007) and the Geological Survey of China (DD20230484). He played a role in writing – review & editing.

**Competing interests:** The authors have declared that no competing interests exist.

both temporal and spatial levels. This approach provides a theoretical foundation for monitoring soil fertility trends and offering rational guidance for production management [4, 5].

In recent years, numerous scholars have made significant advancements in the fields of spatial distribution of soil nutrients, ecological chemical measurement, and comprehensive evaluation [6–9]. For instance, Zheng et al. [10] employed geostatistics, GIS, and fuzzy mathematics methods to analyze the spatial distribution and grade evaluation of soil nutrients in Xuwen County, Guangdong Province. Song et al. [11] explored the C, N, and P contents and ecological stoichiometry of grassland plant-soil-microbial systems under different restoration measures in karst desertification areas. He et al. [12] employed classical statistics, geostatistics, and GIS evaluation methods to investigate the spatial distribution pattern of soil nutrients and comprehensive fertility in medicinal herb planting areas. Recent studies have demonstrated significant advancements in the fields of soil nutrient assessment, particularly through the use of modern techniques such as remote sensing and machine learning. Abdulraheem et al. [13] reviewed the advancements in remote sensing for soil measurements, showcasing the potential of these technologies to enhance soil management practices. Comparative studies in other agricultural regions have provided valuable insights into the effectiveness of various soil assessment methods. These studies highlight the widespread application of spatial statistical methods and GIS in soil nutrient evaluation.

In previous studies, when conducting comprehensive evaluations of soil nutrients using GIS technology, spatial interpolation methods were often employed to generate spatial distribution maps of soil nutrients [14–16]. However, the evaluation results obtained from traditional spatial interpolation methods tend to be overly smooth, resulting in a lower level of precision in local soil nutrient evaluations. This limitation hinders farmers from obtaining more accurate nutrient evaluation results and plot boundary information within their land areas, detrimental to effective scientific fertilization management. A plot is considered the smallest agricultural production and management unit, characterized by high boundary stability. It serves as the fundamental unit for crop production planning, management, and benefits evaluation [17, 18]. Different land uses also have different effects on soil nutrients [19]. In practical agricultural production, using a plot as the smallest unit for soil nutrient evaluation offers several advantages. It allows farmers to quickly and accurately understand the soil nutrient conditions within their land areas. By applying appropriate fertilization practices, farmers can enhance soil fertility, thereby increasing crop yield and income [20]. GIS technology allows researchers to visualize and analyze spatial data, making it an invaluable tool in agricultural research. By using spatial interpolation methods such as kriging and inverse distance weighting (IDW), it is possible to create continuous surface maps that accurately represent soil nutrient levels across a study area. These methods enable researchers to account for spatial variability and provide precise nutrient assessments at the plot level.

Effective soil nutrient management is critical for optimizing crop yields and ensuring sustainable agricultural practices. However, traditional soil assessment methods often lack the precision for targeted fertilization. This study aims to address nutrient management challenges by incorporating plot boundary information into soil nutrient assessments. By providing farmers with more accurate data, the study aims to enhance fertilization strategies, improve crop production efficiency, and contribute to sustainable agricultural practices. This study distinguishes itself by integrating plot boundary information into the spatial interpolation of soil nutrient levels. This novel approach enhances the precision of nutrient assessments, providing a more refined basis for scientific fertilization practices in Kashi Prefecture, Xinjiang. The findings are expected to offer valuable insights for farmers, policymakers, and researchers, advancing the field of precision agriculture and soil nutrient management.

To ensure clarity, key technical terms are defined as follows: Ammonium nitrogen (AN) is a form of nitrogen readily available for plant uptake. Available phosphorus (AP) and available potassium (AK) are essential macronutrients for plant growth, while soil organic matter (SOM) is crucial for maintaining soil structure and fertility. Spatial interpolation refers to the process of estimating unknown values at certain locations based on known values from surrounding locations. GIS technology facilitates the integration and analysis of spatial data.

## Study area and sample

### Overview of the study area

The research area is situated at the western edge of the Kashgar Delta and at the forefront of the Gezi River alluvial fan in the western part of the Taklamakan Desert (Fig 1). It shares borders with Shule County to the northeast, Shufu County to the west, and Aktau County to the south. The research area is centrally located among six counties and cities in Kashgar and Kezhou prefectures. It is approximately 22 km away from the downtown area of Kashgar city. The area's terrain exhibits a high northwest and low southeast pattern, with an elevation ranging from approximately 1272 to 1313 m. The climate in the research area is characterized by low precipitation and high evaporation rates. The average annual temperature is around 12.8°C, the average annual precipitation is 71.6 mm, and the average annual evaporation is 2242.8 mm. The climate in this region is classified as a typical warm, temperate continental arid climate.

### Sample collection and processing

The collection and processing of soil samples in this study adhered to the guidelines outlined in the Land Quality Geochemistry Evaluation Standards (DZ/T0925-2016). The sampling points were determined using a combination of a grid and spot pattern on a land use map. The selection of sampling points was based on the principles of representativeness and evenness. The sampling work was carried out between March and June 2023, resulting in a total of 187 soil samples being collected (refer to Fig 1 for specific locations). The sample density was set at 4 points per square kilometer, with each square kilometer divided into a 2×2 grid. The largest cultivated area within each grid was chosen for sample placement.

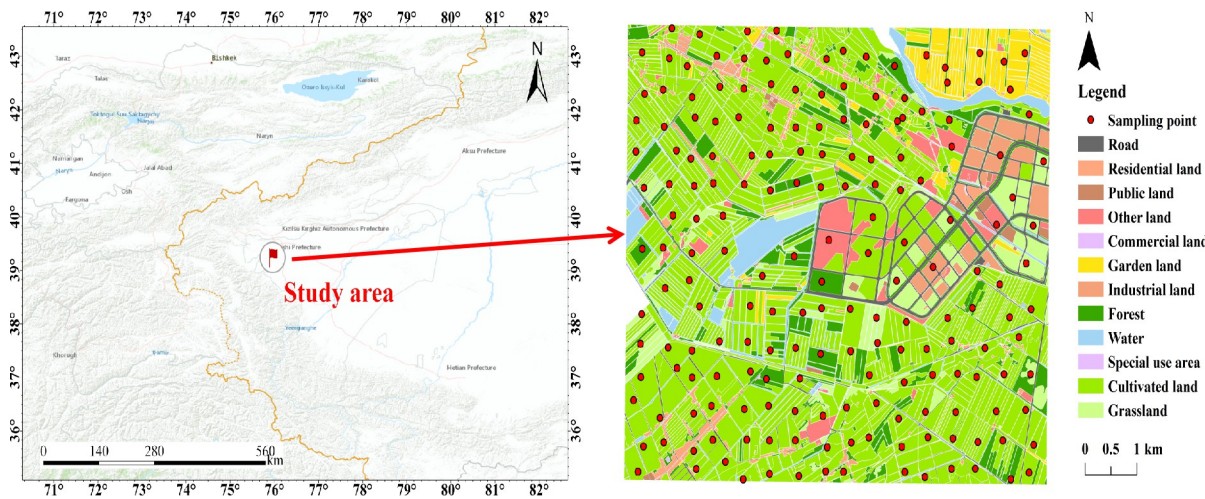

**Fig 1. Summary map of the study area.**

The soil samples were collected in continuous soil columns ranging from 0 cm to 20 cm in depth, with a minimum mass requirement of 1.5 kg. After collection, the samples were naturally dried and then sieved using a 10-mesh screen before being sent to the laboratory for further processing and analysis. The main analysis indicators included total nitrogen (AN), total phosphorus (AP), total potassium (AK), and organic matter (SOM).

Both external and internal quality control methods were employed to ensure the quality of the soil sample analysis and meet the standard requirements for research. External quality control includes testing the quality of laboratory personnel, experimental equipment, and experimental methods by participating in the evaluation of third-party evaluation institutions to ensure the technical level and quality of the laboratory. At the same time, this study invited several professional units in Xinjiang to conduct synchronous measurements of experimental samples and conducted a comparative analysis of the measured results to ensure that the laboratory results are comparable with those of other laboratories. Internal quality control measures are implemented in the laboratory, and repeated measurements are made for the same batch of samples, including parallel sample analysis, standard sample analysis, standard material or quality control sample comparison analysis to ensure the stability and accuracy of the analysis process and results. By implementing these scientific and rigorous sampling and processing measures, the representativeness and reliability of the soil samples were ensured, and accurate analytical results were obtained. These data serve as a solid foundation for soil nutrient assessment and provide reliable support for further research endeavors.

## Methods

The research plan is outlined in Fig 2. The first step involved conducting a spatial interpolation analysis of the soil sample data using GIS. This analysis resulted in creating a spatial distribution map of soil nutrients. In the second step, plot boundary information was introduced to identify the spatial attribute relationship between the soil samples and the plots, allowing for assigning values to the evaluation units. The final step involved comprehensively evaluating soil nutrients using the weighted overlay method. This method considers the assigned values from the previous step and combines them with appropriate weights to comprehensively evaluate soil nutrient levels. Overall, this research plan uses GIS techniques and spatial analysis methods to assess the spatial distribution and comprehensive evaluation of soil nutrients in the study area.

### Soil nutrient single index evaluation for plot unit

1. The evaluation unit in this research is determined to be the plot. This choice is based on the consideration that using a plot as the smallest evaluation unit provides long-term stability and can offer essential evidence for crop production planning, management, and benefit evaluation [21]. By using plots as the basic unit, the soil nutrient evaluation can provide valuable information for agricultural practices.

2. The discrimination of spatial attributes between sampling points and plots is essential in this research. If there is only one data point within a plot unit, that measured data becomes the representative data for that evaluation unit. No other interpolated data within the evaluation unit is considered in this case. However, if there are two or more data points within a plot unit, the average of the measured data is taken as the numerical representation of the evaluation unit. Again, no other interpolated data within the evaluation unit is considered. In situations with no evaluation data available within a plot unit, interpolation methods can be employed to obtain corresponding evaluation data for each evaluation unit. This ensures

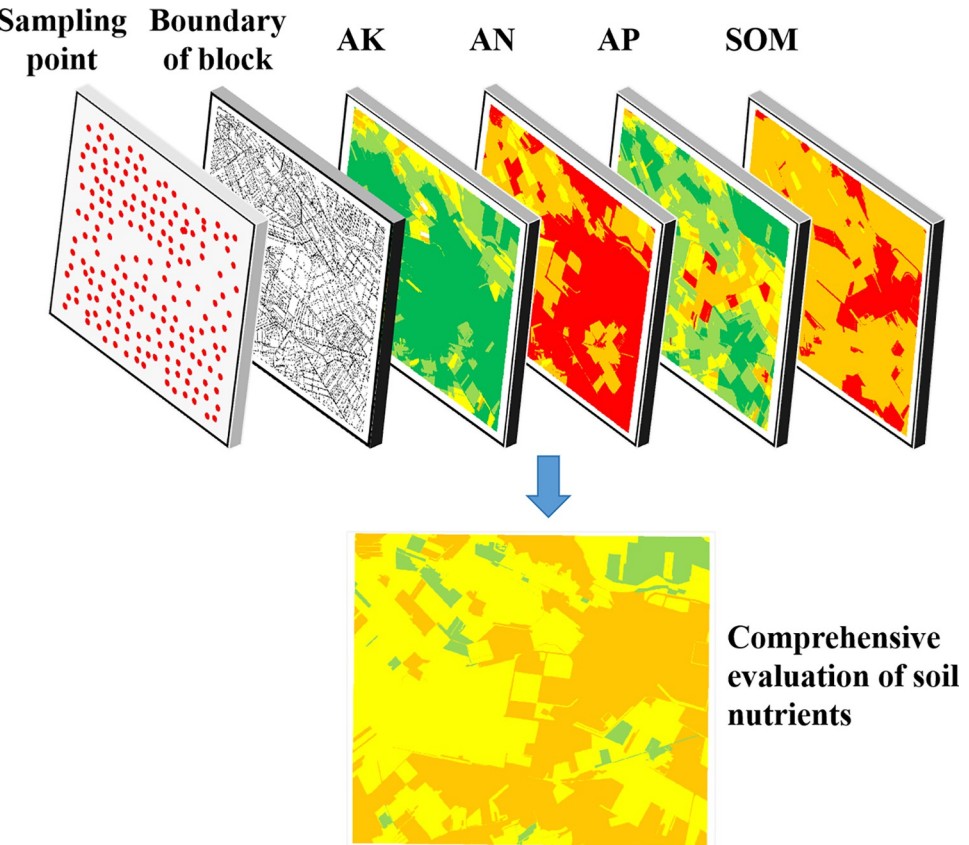

**Fig 2. Schematic diagram of soil nutrient comprehensive evaluation.**

that every plot unit has a representative value for soil nutrient evaluation. By following these steps, the research ensures that each plot unit is accurately represented in the soil nutrient evaluation process, whether through measured data or interpolation methods.

3. To conduct the spatial interpolation of soil nutrients, the research adopts the Kriging method. Kriging is a widely used interpolation method in geostatistics. It is based on a theoretical model of variation function and structural analysis, allowing for unbiased and optimal estimation of variables within local areas [22, 23]. While using Kriging for spatial interpolation, it is essential to determine the semi-variogram function. The semi-variogram function can be calculated using the following formula:

$$\gamma(h) = \frac{1}{2N(h)} \sum_{i=1}^{N(h)} \left[ Z(x_i) - Z(x_i + h) \right]^2 \qquad (1)$$

In the formula, $\gamma(h)$ is the semivariogram function, $h$ is the lag distance or step size, $N(h)$ is the number of samples at a distance equal to h, and $Z(x_i)$ and $Z(x_i+h)$ are the measured values of regionalized variable $Z(x)$ at location $x_i$ and $x_i+h$, respectively. During the spatial interpolation process using Kriging, it is important to select the optimal semivariogram function and fitting model in order to improve the accuracy of interpolation (Kriging method:ordinary; semivariogram model:spherical; search radius:variable; number of points:12).

(4) Evaluation of accuracy: The spatial interpolation results are evaluated using the Root Mean Square Error (RMSE), a commonly used evaluation index for spatial interpolation.

**Table 1. Classification standard of soil nutrient index.**

| Name | Rank I | Rank II | Rank III | Rank IV | Rank V |
|---|---|---|---|---|---|
| AN(g/kg) | >2 | (1.5, 2] | (1, 1.5] | (0.75, 1] | ≤0.75 |
| AP(g/kg) | >0.04 | (0.02, 0.04] | (0.01, 0.02] | (0.005, 0.01] | ≤0.005 |
| AK(g/kg) | >0.2 | (0.15, 0.2] | (0.1, 0.15] | (0.05, 0.1] | ≤0.05 |
| SOM(g/kg) | >40 | (30, 40] | (20, 30] | (10, 20] | ≤10 |
| F | ≥4.5 | [3.5, 4.5) | [2.5, 3.5) | [1.5, 2.5) | <1.5 |

RMSE can quantitatively measure the deviation between predicted data and actual values [24]. A smaller RMSE value indicates that the prediction results are closer to the actual values, indicating higher accuracy in the interpolation. In order to better verify the accuracy of the model, 80% of the data set is used as the training set, and the remaining 20% is used as the training set. The calculation formula of RMSE is as follows:

$$\text{RMSE} = \sqrt{\frac{1}{n}\sum_{i=1}^{n}\left[Z^*(x_i) - Z(x_i)\right]^2} \tag{2}$$

In the formula, $Z^*(x_i)$ is the predicted value of sample xi, and $Z(x_i)$ is the measured value.

## Comprehensive evaluation of soil nutrients based on weight superposition

Based on evaluating soil AN, AP, AK, and SOM individually, referring to the standards of "Standard for Earth Quality Geochemistry Assessment" (DZ/T0925-2016), a comprehensive evaluation of soil nutrient levels is performed (Table 1). According to the requirements of the specification, the comprehensive evaluation criteria are as follows:

$$F = \sum_{i=1}^{n} k_i \times f_i \tag{3}$$

In the formula, $F$ is the comprehensive evaluation level of soil nutrients; $k_i$ is the weight coefficient of soil AN, AP, AK, and SOM; and $f_i$ is the score of the single index level of AN, AP, AK, and SOM. According to the standards of the specification and related research results [25], the weights of AN, AP, AK, and SOM are set to 0.3, 0.3, 0.2, and 0.2, respectively. At the same time, the corresponding $f_i$ scores for Rank V, Rank IV, Rank III, Rank II, and Rank I are 1 point, 2 points, 3 points, 4 points, and 5 points, respectively. We can calculate the comprehensive evaluation level of soil nutrients using this comprehensive evaluation method. The weighted method considers the importance of each nutrient index and the evaluation level score of each nutrient index, thereby obtaining a more comprehensive and accurate evaluation result of soil nutrients [26, 27].

## Results

### Descriptive statistical analysis of soil nutrients

An understanding of the overall changes in soil nutrient levels can be obtained through a descriptive statistical analysis of 187 soil samples collected from the study area. The statistical results presented in Table 2 reveal the mean values of AN (Ammonium Nitrogen), AP (Available Phosphorus), AK (Available Potassium), and SOM (Soil Organic Matter) to be 0.74 g/kg, 0.03 g/kg, 0.33 g/kg, and 13.3 g/kg, respectively. The variation ranges of the soil's AN, AP, AK, and SOM contents are 0.24–1.40 g/kg, 0–0.12 g/kg, 0–2.55 g/kg, and 3.46–26.60 g/kg, respectively. Combining the distribution maps of soil nutrients and violin plots depicted in Fig 3A and 3B allows a more detailed observation of nutrient distribution.

**Table 2. Statistical analysis of soil nutrients.**

| Name | Maximum | Minimum | Average Value | Variance | Standard Deviation |
|---|---|---|---|---|---|
| AN(g/kg) | 1.40 | 0.24 | 0.74 | 0.05 | 0.23 |
| AP(g/kg) | 0.12 | 0 | 0.03 | 0 | 0.02 |
| AK(g/kg) | 2.55 | 0 | 0.33 | 0.16 | 0.40 |
| SOM(g/kg) | 26.6 | 3.46 | 13.30 | 18.20 | 4.26 |

Based on the figures and statistical results, it can be concluded that the AN content is predominantly distributed between 0.4 and 1.0 g/kg, accounting for 85.03% of the samples. Similarly, the AP content is primarily distributed between 0 and 0.05 g/kg, accounting for 84.49% of the samples. The AK content exhibits a distribution mainly between 0 and 0.4 g/kg, encompassing 83.96% of the samples. Furthermore, the SOM content is primarily distributed between 5 and 20 g/kg, accounting for 92.51% of the samples. Regarding standard deviations, AN, AP, and AK display relatively small standard deviations of 0.23, 0.02, and 0.40, respectively, while SOM exhibits a relatively large standard deviation of 4.26.

The normal distributions of AN, AP, AK, and SOM, as depicted in Fig 3C, indicate their suitability for Kriging spatial interpolation, thus satisfying the experimental requirements. By employing the soil nutrient index level classification criteria outlined in Table 1, the following conclusions can be drawn from the statistical analysis: the AN and SOM ratings in the study area fall under rank IV, signifying relatively low nutrient contents, whereas the AK and AP ratings correspond to rank I and II, respectively, indicating relatively high nutrient contents.

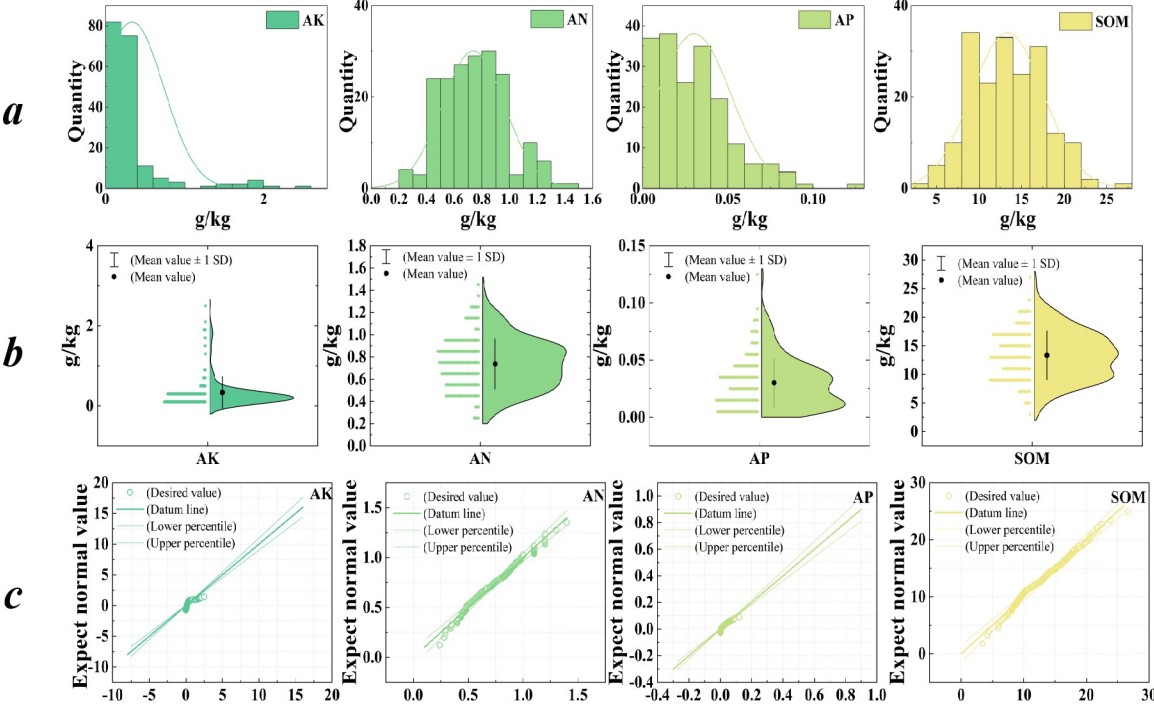

**Fig 3.** Statistical map of soil nutrients: (a) distribution map; (b) Violin diagram; (c) Q-Q chart.

## Spatial analysis of single index of soil nutrients

Fig 4 presents the evaluation results of soil nutrients at individual indexes in the study area. Applying the ordinary Kriging method for spatial interpolation yields root mean square errors (RMSE) of 0.98, 0.93, 0.78, and 0.99 for AN, AP, AK, and SOM, respectively. These values indicate a high level of accuracy in the spatial interpolation results, which effectively meets the research requirements. Regarding the distribution of soil nutrient levels, the AP content

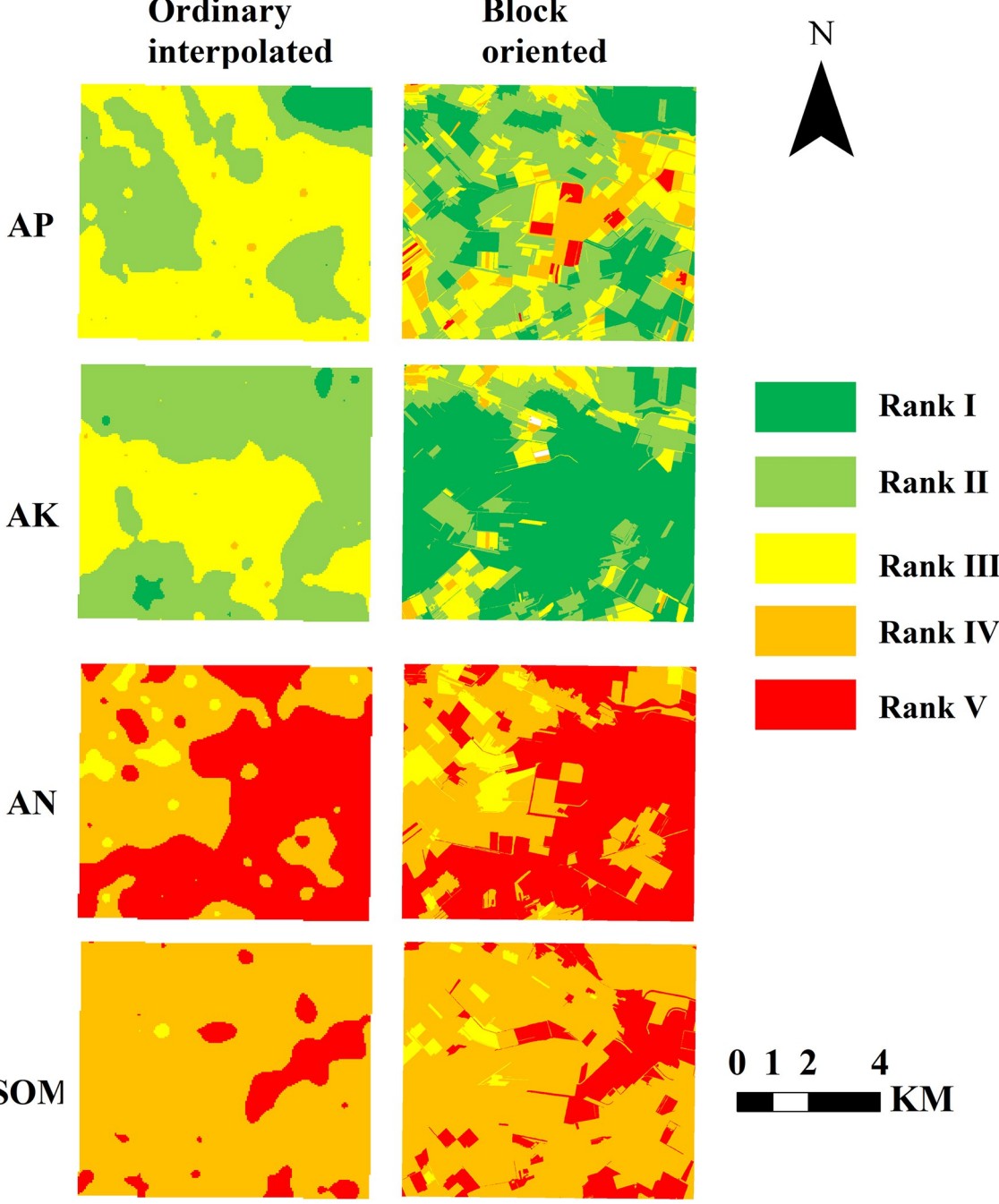

**Fig 4. Results of single index evaluation of soil nutrients.**

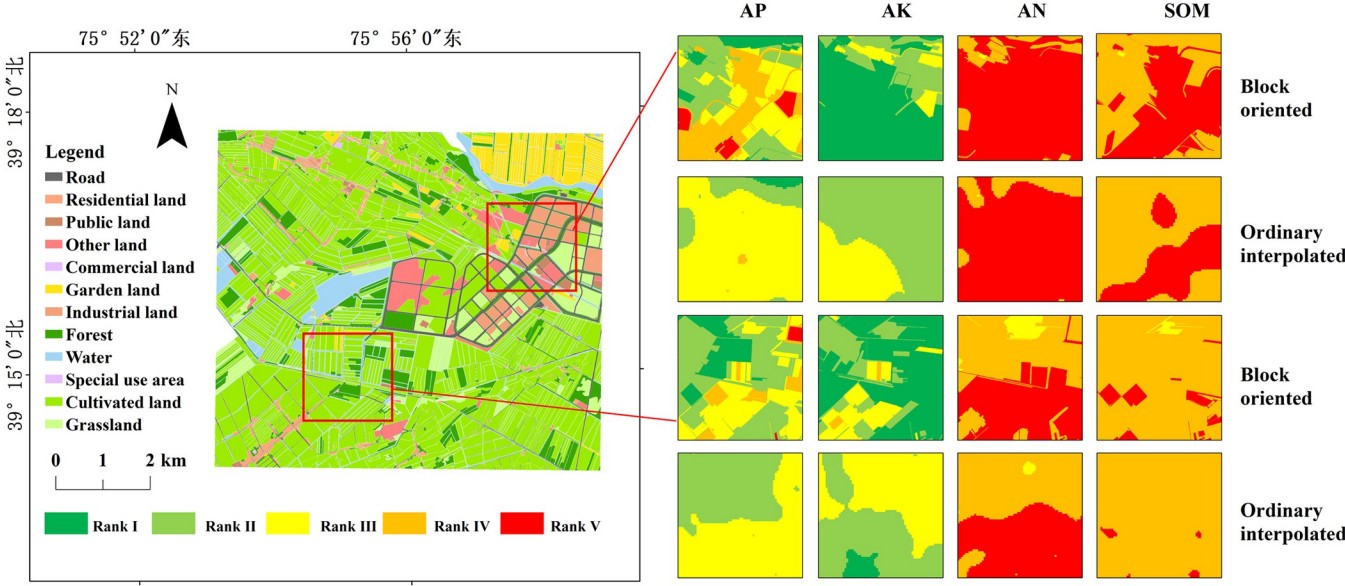

**Fig 5. Local detail map of single index evaluation of soil nutrients.**

predominantly falls within rank II, covering an area of 23.61 km$^2$ and 40.61% of the total area. This is followed by rank I and III, which encompass 13.29 km$^2$ and 12.91 km$^2$, accounting for 22.86% and 22.20% of the total area, respectively. The AK content primarily corresponds to rank I, covering an area of 37.74 km$^2$ and accounting for 64.91% of the total area. Additionally, rank II encompasses an area of 12.16 km$^2$, accounting for 20.92% of the total area. The AN content level is mainly categorized as rank V, covering an area of 32.01 km$^2$ and accounting for 55.07% of the total area. As for SOM, the majority falls within rank IV, with an area of 45 km$^2$, accounting for 77.41% of the total area.

From a spatial perspective, the areas with high AP content are primarily distributed in the study area's southeast, northeast, and northwest regions. The high-value areas for AK content are mainly concentrated in the central part of the study area. On the other hand, the low-value areas for N and SOM content are primarily located in the southeast region of the study area. The western part of the study area exhibits relatively good soil quality, with significantly higher soil nutrient content than the eastern part.

The study compared the advantages of soil nutrient evaluation methods based on plot units with the ordinary interpolation method. Fig 4 shows that the evaluation results of AN, AP, AK, and SOM for plot units are similar to those obtained using the ordinary interpolation method regarding global distribution. The spatial distribution characteristics of high-value and low-value areas are relatively consistent between the two methods. However, Fig 5 reveals some differences between the evaluation method based on plot units and the ordinary interpolation method. The evaluation results of soil nutrients for plot units exhibit more significant boundary characteristics, and the local soil nutrient evaluation levels are more precise. On the other hand, the evaluation results obtained using the ordinary interpolation method are smoother, and the local soil nutrient evaluation levels are rougher.

Farmers are primarily concerned with the soil nutrient conditions within their fields in practical agricultural production. They aim to improve soil fertility through appropriate fertilization practices to increase productivity and income. While the evaluation results obtained using the ordinary interpolation method are smooth and reasonably accurate, farmers may

face challenges in obtaining more intuitive information about the ownership boundaries of their fields and receiving more accurate fertilizer recommendations. Therefore, in practical production applications, evaluating soil nutrients based on plot units achieves high accuracy and helps farmers quickly obtain more intuitive information about the ownership boundaries of their fields and receive more accurate fertilizer recommendations. This approach provides valuable insights for farmers to make informed decisions and optimize their farming practices.

## Comprehensive evaluation of soil nutrients

Fig 6 presents the comprehensive evaluation results of soil nutrients in the study area. Based on the classification criteria for nutrient index levels, most soil nutrients in the study area fall within ranks II, III, and IV. Rank II covers an area of 4.05 km$^2$, accounting for 6.97% of the total area. Rank III encompasses an area of 29.08 km$^2$, accounting for 50.03% of the total area. Rank IV covers an area of 25 km$^2$, accounting for 43% of the total area. These results indicate that most of the plots in the study area have good soil nutrient conditions, providing suitable soil environments for crop growth.

From the spatial distribution characteristics of the comprehensive level of soil nutrients, areas with relatively rich soil nutrients are primarily located in the northeast region of the study area, with sporadic occurrences in the northwest and southeast regions. The soil nutrients in the western and southeastern parts of the study area are moderately abundant, while the soil nutrients in the eastern part are relatively scarce. In addition to considering the soil nutrient situation, other factors such as soil texture and water status should also be considered when assessing crop growth. A comprehensive approach to these factors can help develop more effective soil management and fertilization plans, improving crop production efficiency.

Therefore, to further enhance the overall soil fertility of the region, it is essential to focus on the soil nutrient conditions in the eastern part of the study area. By implementing appropriate fertilization measures and increasing the content of AN and SOM in the soil, the comprehensive level of soil nutrients can be improved, leading to increased crop yields and income. At the same time, more attention should be paid to the content of AP and AK in the central area of the study area, and the soil condition can be further improved by adding nutrients.

## Discussion

Agricultural activities are essential for sustainable development in arid and semiarid regions [28, 29]. Different human activities and land use methods impact the environment differently [30, 31]. The proposed comprehensive evaluation method of soil nutrients for plot units is beneficial for farmers to understand the nutrient situation in their fields and implement scientific fertilization management, ultimately leading to increased crop yields. It also contributes to environmentally, socially and economically sustainable development. The study has achieved positive results, and certain aspects deserve further exploration.

1. In the evaluation method, the weighting and score allocation of AN, AP, AK, and SOM were determined based on the Land Quality Evaluation Standards (DZ/T0925-2016) and related research results. Different weight criteria affect the final soil nutrient evaluation results. In future work, we will explore using an objective weighting method to assign values in different areas to make the evaluation method more generally applicable and reduce subjective bias.

2. In this study, the Kriging method is used for spatial interpolation, and the root mean square error (RMSE) was used to evaluate the spatial interpolation results. The results show that the Kriging method has high interpolation accuracy and can meet the research

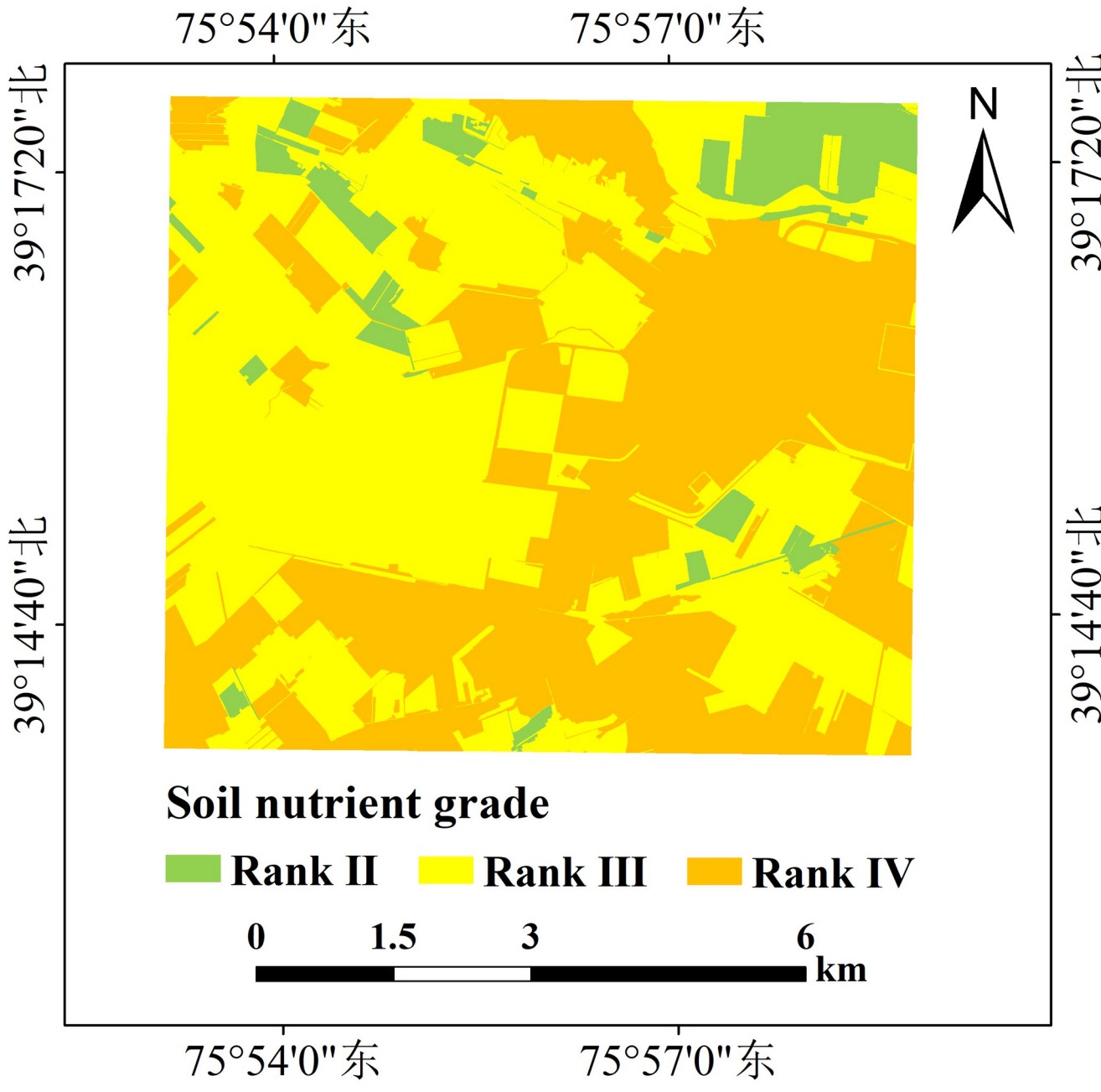

**Fig 6. Soil nutrient comprehensive evaluation map.**

requirements. In the follow-up research, it is suggested to try more spatial interpolation methods to explore the possibility of further improving the accuracy.

3. This study has achieved good experimental results in the Kashgar area of Xinjiang this year. From the principle of model construction, we can know that the model is universal and applicable in different regions. In order to further improve the accuracy and applicability of the model, we will continue to conduct experiments in different regions in the future. Further, explores the usability of the model in different land use modes and environmental conditions. At the same time, the multi-year data will be further integrated into time series

data to analyze the spatio-temporal change characteristics of soil nutrients and establish a time-series evaluation model. This is also more conducive to in-depth analysis of the impact of different nutrients on soil nutrients and establishing the time dynamic weight to improve the model's accuracy.

By addressing these essential points in future research, the comprehensive evaluation method can be further refined and enhanced, leading to more precise and reliable assessments of soil nutrients. This will empower farmers with better decision-making information and improve agricultural productivity.

## Conclusions

This study proposed a comprehensive evaluation method of soil nutrients based on plots. These findings provide valuable insights into the nutrient status of the research area and emphasize the importance of considering soil nutrient management strategies to optimize crop production and maximize agricultural productivity. By providing farmers with more accurate data, the study can enhance fertilization strategies, improve crop production efficiency, and contribute to sustainable agricultural practices. The main conclusions of this study are as follows:

1. The average values of soil AN, AP, AK, and SOM in the research area were 0.74 g/kg, 0.03 g/kg, 0.33 g/kg, and 13.3 g/kg, respectively. AN and SOM ratings fell within the fourth rank, indicating relatively low content, while AK and AP ratings, respectively, belonged to the first and second rank, indicating relatively high content.

2. The overall level of soil nutrients in the research area was determined to be good, but there were certain areas where nutrient contents were insufficient. The comprehensive grades primarily comprised the second, third, and fourth grades, covering 4.05 km$^2$, 29.08 km$^2$, and 25 km$^2$, respectively. These areas accounted for 6.97%, 50.03%, and 43% of the total area.

3. The evaluation results of soil nutrients based on the plot unit exhibited more distinct boundary characteristics and refined rating levels. This is advantageous in meeting the practical production needs of farmers. Therefore, this method is feasible and practical for evaluating soil nutrients.

## Author Contributions

**Conceptualization:** Quan Xu.

**Data curation:** Quan Xu, Junling He.

**Formal analysis:** Quan Xu, Junling He.

**Investigation:** Junling He.

**Methodology:** Quan Xu.

**Project administration:** Junling He.

**Resources:** Junling He.

**Supervision:** Quan Xu, Junling He.

**Validation:** Quan Xu.

**Writing – original draft:** Quan Xu.

**Writing – review & editing:** Junling He.

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
