## [Decision Letter · Decision Letter 0]

22 May 2024

PONE-D-24-18114Refining soil nutrient assessment: incorporating land use boundaries for precision agriculturePLOS ONE

Dear Dr. Xu,

Thank you for submitting your manuscript to PLOS ONE. After careful consideration, we feel that it has merit but does not fully meet PLOS ONE’s publication criteria as it currently stands. Therefore, we invite you to submit a revised version of the manuscript that addresses the points raised during the review process

Below are the comments of three reviewers regarding your submission to PLOS One. The three reviewers have made substantive critical comments, and you should pay close attention to them when making your revisions. The reviewers' comments are important, as they will assist you in making your paper much more interesting to our readers. Please, address properly all the comments of the reviewers. Please, submit your revised manuscript online by using the Editorial Manager system. ==============================

We look forward to receiving your revised manuscript.

Kind regards,

Ahmed M. Saqr, Ph.D.

Academic Editor

PLOS ONE

Journal Requirements:

2. In your Methods section, please provide additional information regarding the permits you obtained for the work. Please ensure you have included the full name of the authority that approved the field site access and, if no permits were required, a brief statement explaining wh

"This research was funded by the Science and Technology Innovation Fund Project of Natural Resources Integrated Survey Command Center (KC20230015, KC20220007), the China Geological Survey Project (DD20230484), the National Natural Science Foundation of China (U2003109) and the Natural Science Foundation of Xinjiang Uygur Autonomous Region (2022D01A149)."            

4. In this instance it seems there may be acceptable restrictions in place that prevent the public sharing of your minimal data. However, in line with our goal of ensuring long-term data availability to all interested researchers, PLOS’ Data Policy states that authors cannot be the sole named individuals responsible for ensuring data access (http://journals.plos.org/plosone/s/data-availability#loc-acceptable-data-sharing-methods).

5. We note that Figure 1 in your submission contain map images which may be copyrighted. All PLOS content is published under the Creative Commons Attribution License (CC BY 4.0), which means that the manuscript, images, and Supporting Information files will be freely available online, and any third party is permitted to access, download, copy, distribute, and use these materials in any way, even commercially, with proper attribution. For these reasons, we cannot publish previously copyrighted maps or satellite images created using proprietary data, such as Google software (Google Maps, Street View, and Earth). For more information, see our copyright guidelines: http://journals.plos.org/plosone/s/licenses-and-copyright.

Please upload the completed Content Permission Form or other proof of granted permissions as an "Other" file with your submission

Reviewers' comments:

Reviewer's Responses to Questions

**Comments to the Author**

1. Is the manuscript technically sound, and do the data support the conclusions?

Reviewer #1: No

Reviewer #2: Yes

Reviewer #3: Yes

2. Has the statistical analysis been performed appropriately and rigorously? 

Reviewer #1: Yes

Reviewer #2: Yes

Reviewer #3: Yes

3. Have the authors made all data underlying the findings in their manuscript fully available?

Reviewer #1: No

Reviewer #2: Yes

Reviewer #3: Yes

4. Is the manuscript presented in an intelligible fashion and written in standard English?

Reviewer #1: Yes

Reviewer #2: Yes

Reviewer #3: Yes

5. Review Comments to the Author

Reviewer #1: Thank you for inviting me to review the paper entitled “Refining soil nutrient assessment: incorporating land use boundaries for precision agriculture”. The topic of the article is interesting. I recommend reconsideration of the paper after addressing the following modifications:

• What is the novelty of this manuscript? It should be clearly mentioned in the last paragraph of the introduction.

• The introduction needed to be expanded and the literature review needed to be updated with up-to-date references about contamination due to land use and its effects on soil nutrients, e.g.: https://doi.org/10.1007/978-3-031-55665-4_2

• Please, replace all the figures with high-resolution ones.

• Figs.1 & 4 & 5 & 6 have not any spatial coordinates. Please, provide them.

• The font of legends is very small. Please, give a suitable font size for legends.

• You should add the limitation of your study at the end of the paper before the conclusion section.

• What is your recommendation for future studies to enhance the applied methodology? You can discuss how to relate your research findings with sustainable development goals (SDGs) to achieve more environmental, economic, and social benefits. References to cite:

https://doi.org/10.1016/j.ejrh.2024.101703

https://doi.org/10.1016/j.gsd.2024.101087

https://doi.org/10.1007/978-981-99-4101-8_27

https://doi.org/10.1007/978-981-99-1381-7_6

Reviewer #2: Review report uploaded as an attachment since it exceeds 20,000 characters. All 4 questions have been answered and explained thoroughly. Additional comments for the author has also been added. The file is named answers to questions.

Reviewer #3: 1. The introduction section provides a good overview of the importance of soil nutrient assessment, but it lacks a comprehensive review of the existing literature on similar methodologies. Could you expand on the existing methods for soil nutrient evaluation at the plot level and highlight the novelty or advantages of your proposed approach compared to those methods?

2. In the sample collection and processing section, you mention that both external and internal quality control methods were employed to ensure the reliability of the soil sample analysis. Could you provide more details on the specific external and internal quality control measures taken, as this would help assess the robustness of your data?

3. The spatial interpolation method used in this study is Kriging. While Kriging is a widely used technique, there are various other interpolation methods available (e.g., Inverse Distance Weighting, Radial Basis Functions, etc.). Could you justify the selection of Kriging over other methods and discuss the potential advantages or limitations of using Kriging for this particular study?

4. In the comprehensive evaluation of soil nutrients section, you mention that the weights for AN, AP, AK, and SOM were set based on standards and related research results. However, the rationale behind the specific weight assignment is not explained. Could you provide more details on how these weights were determined and discuss the potential impact of different weight assignments on the final evaluation results?

5. The research focuses on evaluating soil nutrient levels within a specific study area. Could you discuss the potential applicability and transferability of your proposed method to other regions or crops, considering potential variations in soil conditions, climate, and agricultural practices?

6. The study primarily focuses on the spatial distribution and evaluation of soil nutrients. However, it does not address the temporal aspect of soil nutrient dynamics. Could you discuss the potential implications of considering temporal variations in soil nutrient levels and how your proposed method could be adapted or extended to account for such variations?

7. The comprehensive evaluation of soil nutrients is based on a weighted overlay method. While this method provides a single overall score, it may not capture the potential interactions or trade-offs between different nutrient components. Could you discuss the limitations of this approach and explore alternative methods that could provide a more nuanced understanding of soil nutrient dynamics?

8. The study provides valuable insights into soil nutrient levels and their spatial distribution within the study area. However, it does not address the potential causes or drivers of the observed patterns. Could you discuss potential factors (e.g., land use, management practices, environmental conditions) that may influence the spatial distribution of soil nutrients and how these factors could be incorporated into future research?

6. PLOS authors have the option to publish the peer review history of their article (what does this mean?). If published, this will include your full peer review and any attached files.

Reviewer #1: No

Reviewer #2: Yes

Reviewer #3: No

---

## [Author Response · Author response to Decision Letter 0]

5 Jul 2024

We have revised the whole text one by one according to the opinions of experts. and I would like to express my heartfelt respect and gratitude to the expert for your meticulous and professional revision in your busy schedule, which makes the article more rigorous and further improved. Thanks to the editor and experts for the opportunity to revise. If there is any problem, please feel free to contact me at any time. I am very willing to make positive changes.

---

## [Decision Letter · Decision Letter 1]

24 Jul 2024

Refining soil nutrient assessment: incorporating land use boundaries for precision agriculture

PONE-D-24-18114R1

Dear authors,

We’re pleased to inform you that your manuscript has been judged scientifically suitable for publication and will be formally accepted for publication once it meets all outstanding technical requirements.

Kind regards,

Taimoor Hassan Farooq

Academic Editor

PLOS ONE

Additional Editor Comments (optional):

Dear Authors,

I hope this email finds you well.

I am pleased to inform you that after thorough peer review, your manuscript has been accepted for publication. The reviewers have found your work to be a valuable contribution to the field and have recommended it for publication.

Reviewers' comments:

Reviewer's Responses to Questions

**Comments to the Author**

1. If the authors have adequately addressed your comments raised in a previous round of review and you feel that this manuscript is now acceptable for publication, you may indicate that here to bypass the “Comments to the Author” section, enter your conflict of interest statement in the “Confidential to Editor” section, and submit your "Accept" recommendation.

Reviewer #2: All comments have been addressed

Reviewer #3: All comments have been addressed

2. Is the manuscript technically sound, and do the data support the conclusions?

Reviewer #2: Yes

Reviewer #3: Yes

3. Has the statistical analysis been performed appropriately and rigorously? 

Reviewer #2: Yes

Reviewer #3: Yes

4. Have the authors made all data underlying the findings in their manuscript fully available?

Reviewer #2: Yes

Reviewer #3: Yes

5. Is the manuscript presented in an intelligible fashion and written in standard English?

Reviewer #2: Yes

Reviewer #3: Yes

6. Review Comments to the Author

Reviewer #2: Review uploaded as an attachment since it exceeds 20,000 characters. The name of the file is PlosOne Reviewer Report

Reviewer #3: (No Response)

7. PLOS authors have the option to publish the peer review history of their article (what does this mean?). If published, this will include your full peer review and any attached files.

Reviewer #2: **Yes: **Bright Fafali Dogbey

Reviewer #3: No

---

## [Editor Report · Acceptance letter]

30 Jul 2024

PONE-D-24-18114R1 

PLOS ONE

Dear Dr. Xu, 

I'm pleased to inform you that your manuscript has been deemed suitable for publication in PLOS ONE. Congratulations! Your manuscript is now being handed over to our production team.

Kind regards, 

on behalf of

Taimoor Hassan Farooq 

Academic Editor

PLOS ONE